# Early Detection and Control of the Next Epidemic Wave Using Health Communications: Development of an Artificial Intelligence-Based Tool and Its Validation on COVID-19 Data from the US

**DOI:** 10.3390/ijerph192316023

**Published:** 2022-11-30

**Authors:** Teddy Lazebnik, Svetlana Bunimovich-Mendrazitsky, Shai Ashkenazi, Eugene Levner, Arriel Benis

**Affiliations:** 1Department of Cancer Biology, Cancer Institute, University College London, London WC1E 6DD, UK; 2Department of Mathematics, Faculty of Natural Sciences, Ariel University, Ariel 4070000, Israel; 3Adelson School of Medicine, Ariel University, Ariel 4077625, Israel; 4Department of Applied Mathematics, Faculty of Sciences, Holon Institute of Technology, Holon 5810201, Israel; 5Faculty of Industrial Engineering and Technology Management, Holon Institute of Technology, Holon 5810201, Israel; 6Department of Digital Medical Technologies, Holon Institute of Technology, Holon 5810201, Israel

**Keywords:** social media, online social networking, social factors, health communication, health belief model, health policy, time series, machine learning, epidemiologic methods, computer simulation, epidemics, pandemics, coronavirus, Sars-Cov-2, influenza

## Abstract

Social media networks highly influence on a broad range of global social life, especially in the context of a pandemic. We developed a mathematical model with a computational tool, called EMIT (Epidemic and Media Impact Tool), to detect and control pandemic waves, using mainly topics of relevance on social media networks and pandemic spread. Using EMIT, we analyzed health-related communications on social media networks for early prediction, detection, and control of an outbreak. EMIT is an artificial intelligence-based tool supporting health communication and policy makers decisions. Thus, EMIT, based on historical data, social media trends and disease spread, offers an predictive estimation of the influence of public health interventions such as social media-based communication campaigns. We have validated the EMIT mathematical model on real world data combining COVID-19 pandemic data in the US and social media data from Twitter. EMIT demonstrated a high level of performance in predicting the next epidemiological wave (AUC = 0.909, *F*_1_ = 0.899).

## 1. Introduction

Communication is currently a major strategic aspect of public health management. From the classical epidemiology viewpoint, a clear understanding of the biological factors influencing a disease’s spread is the basis for its detection and control as early as possible. In the context of hazardous events such as pandemics, numerous complex variables influence the pathogen spread, including public behavior. However, nowadays this knowledge must be augmented by including social factors, such as the influence of social media networks (SMNs), as an integral component of the decision tree, to adapt to near real-time public health policies and anticipate disease spread. Artificial intelligence (AI) and machine learning are used daily to support medical practices. We have therefore developed a health communication AI-based tool for early detection and control of an outbreak, which is demonstrated in the context of a current viral pandemic. Epidemic and Media Impact Tool (EMIT) is an AI-based tool based on this approach.

### 1.1. COVID-19 Pandemic Overtime

The coronavirus disease 2019 (COVID-19) pandemic, caused by the severe acute respiratory syndrome coronavirus 2 (SARS-CoV-2), had a major impact on global health, economics, human behavior, and even politics, with an overload on health services, and inducing population anxiety, social distancing, quarantines, and unemployment [1,2,3]. According to the World Health Organization COVID-19 Dashboard, as of 6 November 2022 there have been worldwide 632.6 million confirmed cases and 6.6 million cumulative deaths [4].

Two years into the pandemic, it is clearly evident that the epidemiology of COVID-19 is characterized by recurrent worldwide waves separated by short periods of a very low number of cases [4,5]. The waves are caused by new mutants of SARS-CoV-2 that are produced relatively frequently, as this ssRNA virus has no effective correction mechanisms for gene errors. Some of the natural mutations are advantageous, leading to variants of concerns (VOCs), with alterations in receptor binding, transmissibility, infectivity, severity, and the ability to escape natural infection- and/or vaccine-derived immunity [6,7,8]. The broader population and the higher number of hosts that are infected by the virus, the higher are the chances of the development and spread of VOCs. Since December 2019, four major waves of COVID-19 occurred, which were caused by meaningful lineage mutations: Alpha (B.1.1.7 variant), Beta (B.1.351 variant), Delta (B.1.617.2 variant), and Omicron (B1.1.529 variants), with multiple additional minor variants [5,6].

### 1.2. Clinical Detection of New Pandemic Waves

Detection of a new wave is currently based mainly on reporting of cases by healthcare personnel. This mechanism leads to an inherent delay in apprehension. Since early detection of a new wave is crucial to initiate appropriate measures to prevent its spread and to organize the health system accordingly [5,9], we suggest utilizing health communications and machine learning for earlier detection of a new pandemic wave and assessing the potential effects of certain interventions.This approach follows a wide spread usage of AI-based tools for controlling COVID-19 spread [10,11,12,13,14].

### 1.3. Health Communications-Based Detection of New Waves

SMNs currently play a predominant role in the overall mass communication landscape [15]. Therefore, in the context of epidemics and pandemics, changes in the disease rate or its patterns are immediately reflected in these networks. Indeed, every individual who witnesses a change in disease manifestations or rates - anyone, anywhere, and at any time - can post messages that are readable and easily shareable by millions. People use the SMNs to broadly share their thoughts regarding recommended medical treatments and preventive measures, including vaccines, which have a major impact on the population. It has been documented, for example, that SMNs are common sources of information on measles and the vaccine against it, and are associated with inaccurate (or even incorrect) knowledge, leading to vaccination hesitancy [16,17]. Similar behavior and infodemics exist for the seasonal influenza vaccination and more recently for the COVID-19 pandemic. It is crucial that healthcare professionals and decision-makers should be aware of this prevailing behavior and respond accordingly on these platforms, with the aid of experts in health communication and social networking [18,19,20]. It is therefore obvious that judicious analysis of public communications in the SMNs is currently an important pillar for early prediction of outbreaks and epidemic waves and their management, including responding to health communications [21].

### 1.4. Aims and Objectives

Our goal is to support health policymakers in the early detection of an epidemic or its next wave and enable early interventions by applying machine learning tools on topics of interest used in SMNs.

The main aim of our research is to build a generic prediction model to enable the forecast of epidemic waves by analyzing topics of relevance in SMNs, including at the sub-population levels (e.g., taking into account socio-demographic characteristics).

Our objective is to demonstrate that the combination of historical data on social media activity and the pandemic spread helps to predict in advance the beginning of the next epidemiological wave, and thus help the decision-makers to enhance interventions to reduce the spread of the disease and its impacts.

## 2. Material and Methods

We develop a tool, called EMIT (Epidemic and Media Impact Tool), that:1predicts the next epidemiological wave (Section 2.2), and2estimates the impact of social interactions on the population’s compliance by using various Pandemic Intervention Policies (PIPs).

### 2.1. Data Collection

We collected 2,782,720 tweets (from 420,617 unique users) from Twitter (using the Twitter streaming application programming interface [22]), all from the United States (US) between 30 December 2019, and 30 April 2021. All tweets are in English. Each tweet contains the message itself and the date of publication alongside other meta-data which has been neglected in the scope of this model. In particular, all the tweets contained at least one of the tokens “flu,” “vaccination”, “vaccine”, and “vaxx”. These terms were selected originally by [23] to maximize the chance to retrieve discussions concerning a vaccine as a product, vaccination as an act or a policy, vaccination hesitancy, and influenza. In addition, we gather the daily number of infected and dead (due to the pandemic) individuals in the same period from the World Health Organization (WHO).

### 2.2. Next Wave Predictor

#### 2.2.1. Overview

We develop an approach to predict pandemic spread, focusing on the wave structure of pandemics [24] and the influence of the willingness of the population to follow PIPs [25]. We use historical data on social media activity and the pandemic spread to predict in advance the beginning of the next epidemiological wave. Given such a wave is predicted at some point, we explore the usage of publishing in social media pro-PIP agenda-leading ads. These aids influence the position of individuals in the population regarding several PIPs. By compiling with PIP, the population will reduce the pandemic spread and therefore accomplish the objective of this approach.

Thus, we proposed a two-fold model with a *next wave*, time-series predictor, with a social-epidemiological simulator of the pandemic spread. The proposed approach is divided into training and inference phases. As part of the training phase, the machine learning (ML)-based *next wave* predictor is obtained as well as a fitting of the social-epidemiological parameters used by the simulator. Later, during the inference phase, the trained *next wave* predictor is obtained from the latest social and epidemiological data and predicted if the next wave will occur in a pre-given delay. If a next wave is predicted to occur, the simulator is computing a baseline prediction of the pandemic spread based on the latest available data about compliance with PIPs, social and physical connections’ topology in the population, and the epidemiological state of the population. Later, the user defines the strategy of the proposed social-PIP and a budget. Given these inputs, the model computed a modified simulation and reports the differences between the baseline and modified simulation results.

#### 2.2.2. Machine Learning Process

The *next wave* sub-model uses natural language processing (NLP) to extract time-series data from social media posts on Twitter and combine it with the time-series signal of the epidemiological data from the WHO.

**Data Pre-Processing** The data have been cleaned and lemmatized (using the Python Natural Language Toolkit) of similar words appearing in posts as well as removing punctuation marks, mentions of users, glyphs, website addresses, and stop words. Moreover, the frequent representations of the term “COVID” were replaced with the single form “covid,” and the terms related to “influenza” were lemmatized to “flu” as the popular language used on Twitter [23].

Once the data is cleaned, we used the classification of topics and clustering methods following the N-gram work embedding [23]. This results in an aggregated number of mentions of each topic on a daily basis. We combine this signal with the epidemiological signal to get the time-series data-set used from this point forward. Finally, we normalize all features independently using the Z-score normalization (i.e., $x\to \frac{x-E[x]}{SD[x]}$).

As part of the model configuration, a delay between the source (x) values and the target (y) values, and the number of days in source values are given. Based on the provided configuration, the data-set is transformed from time series to a regression one.

**Model selection and hyperparameter tuning** We used a combination of a grid search [26] and Tree-Based Pipeline Optimization Tool (TPOT) [27] to find the optimal auto-ML configuration, including the optimal number of days in the source values, model selection, and the model’s hyperparameters. The models used by the TPOT framework are all the regression ML models found in the scikit-learn library [28] and the XGboost model [29]. A schematic view of the model auto-ML training process is shown in Figure 1.

To be exact, we utilize the genetic algorithm approach [30] to find the optimal ML learning pipeline where the fitness (i.e., loss function) of each candidate is computed to be the mean of a k-fold cross validation [31]. Each pipeline can contain a filter feature selection method and an ensemble of regression ML models [27]. During training, the dataset is divided randomly into *training* and *validation* cohorts such that the first contains 80% of the data and the latter contains the remaining 20%.

**Model evaluation** To evaluate the model, the k-fold cross-validation approach with (k=5) has been used with the mean absolute error (MAE) metric over the number of infected individuals. Afterward, we utilize the “next wave” classification model on the target (y) variable to obtain a binary vector. The binary definition of the beginning of a “next wave” is obtained by computing the majority vote between five domain experts from the WHO, American Health Association, and European Union Health Committee. In a similar manner, we computed k-fold cross-validation (k=5) over the recall, precision, accuracy, and area under the receiver operating characteristic (ROC) curve (AUC).

### 2.3. Social-Epidemiological Simulator

The social-epidemiological simulator is based on the classical *SIR* model proposed by [32] and extends it by introducing graph-based spatial dynamics and additional four epidemiological states. Moreover, we extended the *SIR*-type model to a *SIRS*-type model [33,34] which is known to better fit COVID-19 [35,36]. On top of this model, a social PIP which is based on mask-wearing, social distance, and vaccination PIPs is defined.

#### 2.3.1. Social-Epidemiological Dynamics

We define a spatio-temporal sub-model that captures the changes in the epidemiological states of a population as they interact over both epidemiological (physical) and social (virtual) domains. Formally, the sub-model is constricted from a tuple (G,P) where $G=(V,E_{e},E_{s})$ is an underacted, connected, non-empty, graph of the individuals and the interactions between them such that the nodes in the graph (V∈G) corresponding to the individuals in the population p∈P and $E_{e}\subset V\times V$ are the possible epidemiological interactions between the individuals in the population, and $E_{s}\subset V\times V$ are the social interactions between the individuals in the population. *P* is the population of individuals.

The model considers a constant population *P* with a fixed number of individuals |P|=|V|=N. In this model, we neglect the population growth over time due to the relatively short time horizon of interest (up to a few weeks). Each individual p∈P is defined by a timed finite state machine as follows: $p:=(\alpha ,\tau ,\bar{\mu },\bar{\mu },\bar{\omega },\bar{\kappa }$) where $\alpha \in \{S,E,I^{s},I^{a},R^{p},R^{f},D\}$ is the current epidemiological state of the individual, τ∈N is the time passed from the last change of the epidemiological state (e), $\bar{\mu }\in R^{k}$ is a vector of personality properties, such that k∈N is the number of personality properties, $\bar{\omega }\in {\left[0,1\right]}^{l}$ is the vector of wiliness to perform a PIP, such that l∈N is the number of PIPs, and $\bar{\kappa }\in N^{7}$ is the vector that counts how many times the individual was in each epidemiological state.

Individuals were categorized to one of seven epidemiological groups (as indicated by their α parameter): susceptible (S), exposed (E) asymptomatic infected $(I^{a}$), symptomatic infected $\left(I^{s}\right)$, partially recovered $(R^{p}$), fully recovered $\left(R^{f}\right)$, and dead (D) such that $N=S+E+I^{a}+I^{s}+R^{p}+R^{f}+D$. Individuals in the first (susceptible) group have no immunity and are susceptible to infection by the pathogen. When an individual in the susceptible group (*S*) is exposed to the pathogen, the individual is transferred to the exposed (E) at a rate β. Individuals in the exposed group have the pathogen but are not contagious. The individual stays in the exposed group on average ϕ days, after which the individual is transferred to either the asymptomatic infected $\left(I^{a}\right)$ or symptomatic infected $\left(I^{a}\right)$ group, which makes them contagious to others with a rate η and 1−η, respectively. Asymptomatic infected transfer to the fully recovered $\left(R^{f}\right)$ group after $\gamma^{a}$ days. A rate of $\psi_{1}$, $\psi_{2}$ and $\phi_{3}$ of the symptomatic infected individuals are transferred to the partially recovered ($R^{p}$), fully recovered ($R^{f}$), and the dead (D) groups after averaging of $\gamma^{s}$ days, respectively, such that $\phi_{1}+\phi_{2}+\phi_{3}=1\wedge 0\leq {\left\{\phi_{i}\right\}}_{i=1}^{3}$. The partially and fully recovered individuals are again healthy and no longer contagious. However, partially recovered individuals suffer from long term medical problems due to the diseases. The partially and fully recovered become susceptible again at a rate $\chi^{p}$ and $\chi^{f}$ on average, respectively, due to a natural reduction in the antibody level generated by the immune system of each individual. A schematic transition between the epidemiological stages of an individual is shown in Figure 2.

The model assumes a discrete clock that all the individuals in the population follow. At each clock tick, in a random order, both social and epidemiological processes occur. First, without loss of generality, for the social process, each individual in the population (p∈P) updates its wiliness vector ($\bar{\omega }$) as follows [37]:(1)$${\bar{\omega }}_{p}^{t+0.5}={\bar{\omega }}_{p}^{t}+\frac{\lambda_{1}}{|N_{s}\left(p\right)|}\sum_{i\in N_{s}\left(p\right)}F\left({\bar{\omega }}_{p}^{t},{\bar{\omega }}_{i}^{t},{\bar{\mu }}_{p},{\bar{\mu }}_{i}\right)\cdot cosine\left({\bar{\mu }}_{p},{\bar{\mu }}_{i}\right){\bar{\omega }}_{i}^{t},$$
where $\lambda_{1}\in \left[0,1\right]$ is a social influence parameter, $N_{s}\left(p\right)$ is a function that gets an individual (*p*) and returns the group $\left\{i\in P\;|\;e_{s}=\left(p,i\right)\in E_{s}\wedge i\notin D\right\}$, and $F({\bar{\omega }}_{p}^{t},{\bar{\omega }}_{i}^{t},{\bar{\mu }}_{p},{\bar{\mu }}_{i})$ is defined as follows:$$F\left({\bar{\omega }}_{p}^{t},{\bar{\omega }}_{i}^{t},{\bar{\mu }}_{p},{\bar{\mu }}_{i}\right):=\left\{\begin{array}{c} 1,\;cosine\left({\bar{\omega }}_{p}^{t},{\bar{\omega }}_{i}^{t}\right)\leq \epsilon_{\omega }\wedge cosine\left({\bar{\omega }}_{p}^{t},{\bar{\omega }}_{i}^{t}\right)\leq \epsilon_{\mu }\\ -1,\;\left(cosine\left({\bar{\omega }}_{p}^{t},{\bar{\omega }}_{i}^{t}\right)\leq \epsilon_{\omega }\wedge cosine\left({\bar{\omega }}_{p}^{t},{\bar{\omega }}_{i}^{t}\right)>\epsilon_{\mu }\right)\vee \left(cosine\left({\bar{\omega }}_{p}^{t},{\bar{\omega }}_{i}^{t}\right)>\epsilon_{\omega }\wedge cosine\left({\bar{\omega }}_{p}^{t},{\bar{\omega }}_{i}^{t}\right)\leq \epsilon_{\mu }\right)\\ 0,\;cosine\left({\bar{\omega }}_{p}^{t},{\bar{\omega }}_{i}^{t}\right)>\epsilon_{\omega }\wedge cosine\left({\bar{\omega }}_{p}^{t},{\bar{\omega }}_{i}^{t}\right)>\epsilon_{\mu } \end{array}\right.$$
such that $\epsilon_{\omega },\epsilon_{\mu }\in R^{+}$ are threshold parameters that indicate the similarity in personality and willingness that two individuals should share so one individual will accept (or reject) the other’s willingness, respectively. Following that, as individuals in the social group of an individual *p*, $N_{s}\left(p\right)$, are affected by the pandemic in verse negative forms such as symptomatic infected, partially recovered, or dead the individual gets a motivation to follow PIPs with a preoperative of the harm others in its social group experienced:(2)$${\bar{\omega }}_{p}^{t+1}={\bar{\omega }}_{p}^{t+0.5}+\lambda_{2}\sum_{i\in N_{s}\left(p\right)}G\left(i\right),$$
where $\lambda_{2}\in \left[0,1\right]$ is a social influence parameter and G(i) returns a non-negative values ${\left\{d_{j}\right\}}_{j=1}^{3}$ corresponding to the motivation to follow PIPs due to the transformation of individual *i* to the symptomatic infected, partially recovered, and dead epidemiological groups, respectively, such that $d_{k}>d_{j}↔k>j$. At any other case, G(i) returns 0.

Afterward, each infected individual p∈P has a probability β∈[0,1] to infected susceptible individual i∈P if and only if $\left(p,i\right)\in E_{e}$, in a pair-wise manner. If a susceptible individual is infected, it immediately becomes exposed (i.e., α←E). Exposed individuals transform to the asymptomatic or symptomatic infected state where τ=ϕ∈N. Infected individuals transform to either the dead, partially recovered, or fully recovered epidemiological groups where $\tau =\gamma^{a}\in N$ and $\tau =\gamma^{s}\in N$, respectively. Finally, partially and fully recovered individuals transform to the susceptible group where $\tau =\chi^{p}\in N$ and $\tau =\chi^{f}\in N$, respectively.

#### 2.3.2. The Social-PIP

In order to control the pandemic, at the beginning of the dynamics, a set of PIPs are defined, corresponding to the number of willingness’ subjects the individuals have. We assume individuals with a willing score higher than C∈[0,1] for a given PIP will execute this PIP in the next step in time. In particular, we define three types of PIPs: mask-wearing, social distancing, and vaccination.

The mask-wearing PIP reduces the infection rate β by a factor $x_{m}^{1}\in \left[0,1\right]$ if the susceptible individual in the interaction wearing a mask, $x_{m}^{1}\leq x_{m}^{2}\in \left[0,1\right]$ if the infected individual wearing the mask and $x_{m}^{2}\leq x_{m}^{3}\in \left[0,1\right]$ if both individuals wearing masks. The social distance PIP reduces the infection rate β by a factor $x_{s}$ if at least one individual in the interaction preserves the PIP. The vaccination PIP is depended on the number of vaccines an individual obtained so far. Namely, an individual can be vaccinated up to *Z* times, so that between every two vaccinations a duration of θ∈N or more must pass. As other PIPs, the vaccination PIP reduces the infection rate β by a factor H(z,t) if the susceptible individual in the interaction is vaccinated, such that H(z,t) is a continuous, differential function that satisfies:$$\forall t\in R^{+}:\frac{\partial H(z,t)}{\partial z}\geq 0\wedge \forall z\in \left[0,\ldots ,Z\right]:\frac{\partial H(z,t)}{\partial t}\leq 0.$$

A central planner (CP) can influence the individuals’ level of willingness by introducing to the graph, *G*, *virtual individuals* these are an abstract representation of media influence on the individuals such as social media ads, television news, and ads, and similar communication means. We define such virtual individuals as follows: $v:=(\bar{\mu },\bar{\omega })$ where $\bar{\mu }$ and $\bar{\omega }$ defined as shown in Section 2.3.

Formally, the CP has a budget $B\in R^{+}$. The CP can define and buy virtual individual *a* such that each edge $e\in E_{s}$ added to *G* between an individual and virtual individual is associated with a fixed cost *c*. In addition, the CP has a utility function *U* that reflects the pandemic spread. As such, the CP is faced with an optimization problem in the form:(3)$$\min_{E^{*}\subset V\times A}\Sigma_{t=0}^{T}U\left(t\right)\;s.t.\;c\cdot \left|E^{*}\right|\leq B,$$
where $E^{*}$ is the set of social edges between individuals (p∈P) and virtual individuals (a∈A), and T∈N is the duration of time of interest to evaluate the policies performance. A schematic view of the dynamics and epidemiological-social PIPs is shown in Figure 3.

#### 2.3.3. Social-PIP Strategies

Given a budget *B*, a model *M*, and a cost of connection between individual and virtual individual *c*, there are B/c2N for only a single virtual agent. Therefore, a CP is facing an ultra-exponential search space to the size of the population. As such, naively finding the optimal allocation problem is hard. However, one can utilize several strategies with heuristics that aim to obtain a close-to-optimal performance. First, a CP can influence a random subset of individuals each time. This strategy is used as a baseline to analyze the performance of the following strategies. Secondly, a CP can aim to influence *Opinion leaders* which are the individuals in the population with the highest number of social connections. In other words, $OL_{k}:=argmax_{P^{*}\subset P}\left(\sum_{p\in P^{*}}N_{s}\left(p\right)\right)$ such that $|P^{*}|=k$. The motivation for this strategy is that the most connected individuals would spread the pro-PIP stand to the remaining population the fastest. Third, a CP can aim to influence *Anti-individuals* which are the individuals with the lowest will to comply with one or more PIPs of interest. For example, $AI_{k}:=argmin_{P^{*}\subset P}(\sum_{p\in P^{*}}|\bar{\omega }\left|\right)$ such that $|P^{*}|=k$. The motivation for this strategy is that by improving the position of the individuals with the lowest willingness to comply with PIPs, they will spread their anti-agenda about the PIPs to the remaining population. Finally, the *Optimal* strategy can be computed for each signal step in time independently. Formally, one can define a utility function of the social PIP to be the increment in the willingness of the population to comply with the PIPs. Therefore, the utility function takes the form:(4)$$U_{t}\left(P\right):=\sum_{p\in P}\left|\right|\omega_{p}^{t+1}-\omega_{p}^{t}\left|\right|.$$

Since there is no cost for creating virtual individuals, each connection defined by the strategy would be between an individual *p* and a virtual individual vp such that $vp:=argmax_{\omega_{vp},\mu_{vp}}F\left({\bar{\omega }}_{p}^{t},{\bar{\omega }}_{vo}^{t},{\bar{\mu }}_{p},{\bar{\mu }}_{vp}\right)\cdot cosine\left({\bar{\mu }}_{p},{\bar{\mu }}_{vp}\right){\bar{\omega }}_{vp}^{t}$. Now, the strategy can pick either pick or not pick an individual. One can solve this problem using a genetic algorithm [30] with the tournaments selection operator [38]. In particular, we would require that a portion ζ∈(0,1) of the population with the best fitting function is left untouched over generations. Therefore, since the fitness function is linear and configuration space is finite, according to the Simplex theorem an optimal solution exists and will be obtained at some point during random search such as the one utilized by the genetic algorithm [39].

Of note, the *Opinion leaders* and *Anti-individuals* are designed based on the common topology of social networks in nature [40,41,42]. As such, one can artificially design a network with a topology in which these strategies are performing on average worse than a random sample of the population. For instance, for the *Opinion leaders* strategy, one can define a graph *G* with two components, one of size *k* and another with size N−k such that the *k*-sized component is fully connected and each node in the N−k-component has less than *k* social edges. Only a single edge is connecting the two components. It is clear to see that the *Opinion leaders* strategy will pick all the nodes in the *k*-sized component. For N>>k, this strategy would obtain poor results.

#### 2.3.4. Fitting Procedure

We used the fitting procedure proposed in [35] with the epidemiological data from WHO [4] for the US between 30 December 2019 and 30 April 2021. Specifically, the daily number of infected, recovered, and deceased individuals have been used.

## 3. Results

The outcome of our present study is the development of EMIT, which is able to:predict with a high level of accuracy the beginning of a new epidemiological wave;simulate the impact of online promotion of various Pandemic Intervention Policies on the pandemic spread.

### 3.1. EMIT: Epidemic and Media Impact Tool

Health policymakers must be prepared for future epidemiological waves of the current COVID-19 pandemic, as well as for other pandemics. Thus, it is crucial to be able to predict the next wave as soon as possible. Moreover, reducing the impact of the wave on the mass population requires efficiently targeted communication campaigns. To assist decisions taken by health system policymakers, EMIT aims to anticipate and enhance the healthcare systems and population preparedness for infectious diseases spreading and epidemic waves. An additional pillar of this tool consists of analyzing the potential impact of public health interventions to estimate beforehand the most efficient action to be taken. EMIT supports the decision process by detecting and analyzing in a systematic way real-time electronic big data, that reflect an extensive number of population interests and behaviors. The EMIT development core team built an international community to enable other specialists to be involved in the expansion of EMIT capabilities. Therefore, EMIT has been developed in Python, one of the most popular programming languages [43], and is hosted on GitHub (a platform for software development and sharing). A schematic view of the epidemiological wave predictor training process and the social PIP’s evaluation is shown in Figure 4.

### 3.2. Prediction of the Next Epidemic Wave

Figure 5a,b present the official daily number of COVID-19-infected individuals in the USA from the WHO database [44] and the normalized amount of social posts on Twitter that mention influenza, COVID, and Vaccine between 30 December 2019, and 30 April 2021, respectively, [23]. In addition, the horizontal lines indicate the beginning of a new pandemic wave based on:a public health retrospective viewpoint (solid red line), looking at the virus spread also recognizable on Figure 5 as increasing trends and as reported by the authorities [44],a patient-centered viewpoint, looking at clinical symptoms in the community (solid blue line) [44],a predictive modeling approach (dotted-green) taking into account changes in topic trends on Twitter (potentially in any other social media).

This graphical representation combines data related to different COVID-19 waves as detected by the healthcare systems on the one hand, and trends of social media topics, on the other hand. It is possible to see that in the case of this pandemic, SMNs support the prediction of new pandemic waves by indicating some modification in the content and focus of threads among the users of social media. Thus, this should help to build social media-based public health interventions to restrain a new virus spread wave.

In addition, at the beginning of the pandemic, which is the time of the sampled data, the news compared COVID-19 with Influenza. As such, one can notice that the “Covid” and “Influenza” signals are closely related, as shown in Figure 5b.

In particular, for the case of COVID-19 in the US, EMIT obtain an area under the receiver operating characteristic (ROC) curve (AUC) of 0.909 and $F_{1}$ score of 0.899 using data from 30 December 2019, and 30 April 2021.

### 3.3. Simulation of Social Media Network Intervention Policies

Based on the results above, we have simulated, on different population sizes (from a few hundred thousand to dozens of millions), the impact of PIPs on SMNs focusing deferentially on social distancing, mask-wearing, vaccination, and all of these combined. The main objective of this simulation is to highlight the need to define public health intervention policies targeting social media users.

The results of this simulation, as shown in Figure 6, indicates that the different PIPs have an increased efficiency proportional to the time-to-new wave. In other words, the PIPs have an effect proportional to the timing of the prediction of the next wave. Additionally, the values generated by the simulation model indicate that having different strategies for running targeted communication campaigns on SMNs impacts positively and significantly on the targeted sub-populations. Indeed, the ability to anticipate the virus dissemination ($R_{0}$) depends not only on its biological characteristics but also on social and communication behaviors. These last ones are reflected by the population engagement in respecting the PIPs (i.e., social distancing, mask-wearing, and vaccination). Our model deals with four types of social media targeted strategies.

The first aims to be very broad and to communicate on the PIPs with a random group of individuals in the simulated population, without considering their willingness to be engaged in the PIPs or their influence on the social networks concerning the number of “friends” (i.e., other SMNs users that they are connected with). This type of approach seems less efficient for limiting the virus from spreading in any size population.

The second strategy focuses on the “opposing” group that is vaccine-hesitant and has a low willingness to comply with the PIPs. Our model suggests that the communication campaigns targeting this sub-population have some positive effects over time. Indeed, not all these individuals are simultaneously anti-vaccines, anti-social distancing, or anti-mask-wearing. Therefore, targeted communication campaigns can influence some of them to be compliant with at least a component of the PIPs.

The third strategy consists of targeting social media community leaders. Considering that these individuals have a high number of connections in the network and so a high number of followers are likely to be influenced by them. This is based on the situation in which the leaders share messages encouraging the population to be engaged in disease-spreading preventive measures. The motivation for this strategy is that the most connected individuals are likely to spread the pro-PIP position to the remaining population relatively fast. Nevertheless, considering the parameters used for the simulation of the impacts of PIPs, this strategy has a similar impact to the strategy that targets the groups that are hostile to the PIPs.

The last strategy designated “optimal”, consists of targeting specific individuals, based on a genetic algorithm. At each step of the simulation (e.g., each simulated day), individuals are targeted to be included in social media interventions according to the disease-vulnerability status (susceptible, vaccinated and/or wearing a mask and/or practicing social distancing, exposed, infected, recovered, dead) of their connections. According to our model, this approach is the most efficient during the two weeks before the new waves both when each PIP is considered alone or all together.

In summary, the higher likelihood that the PIPs messages are received by the population, the greater will be the population’s response and pro-activeness to the PIPs.

## 4. Discussion

Health communication is currently an essential pillar in public health management. SMNs are nowadays extensively used; they represent a major avenue for the population for searching health-related information, and also a platform for reflecting their understanding, beliefs, and positions regarding health issues. However, although public behavior plays a major role the endemic, epidemic, and pandemic evolution, disease spread is still mostly analyzed from a biological perspective.

It is therefore essential to be able to adapt near real-time public health communication policies to stem threads that can induce a disease spread. Moreover, as healthcare-related SMNs reflect real-world and real-time events, their collection from defined populations and sub-populations, with appropriate analyses, can be an important tool for anticipating disease proliferation. More globally, individuals in the population share "news" on social media networks as well as consume it. This feedback loop can explain why social media can be a source for predicting an epidemiological process. We propose EMIT, a tool that uses machine learning methodologies for analyzing internet search-based data, to support the early detection and control of outbreaks.

EMIT capabilities have been documented in the context of the COVID-19 pandemic. EMIT can accurately predict a new epidemiological wave, and can also support healthcare policymakers in defining and updating PIPs by giving a simulated evaluation of the impact of PIPs promotion on the internet and particularly in the SMNs.

The prediction capabilities of EMIT are based on natural language processing of social data from Twitter [22] and epidemiological data from WHO [4]. These data are used to train a machine-learning-based time-series prediction model to warn about a potential approach to a new pandemic wave.

In addition, complementary prediction of the impact of several interventions is given, based on an extended-*SIR* compartmental model [32] using graph-based spatial dynamics [45,46,47,48] for estimating the effects of PIPs on population activities [49].

Like any computational tool, EMIT has the capability to enhance and improve policymakers’ decision processes. Indeed, EMIT has shown a high accuracy level for predicting the next pandemic wave of AUC=0.909 and $F_{1}=0.899$ scores for a prediction of two weeks ahead of time.

Moreover, EMIT simulation component enables evaluating the effects of potential PIPs before implementing them, by estimating the impact of various isolated or combined counter-measures against disease spread. This crucial information should be taken into account during the process of policy development in order to devise the most efficient PIPs with regard to population behavior [50,51,52].

One additional strength of EMIT is that non-AI health-domain experts get user-friendly and easily understandable data by visualizations of the prediction and simulation computations. Hence, the EMIT’s outputs can be used by health policymakers in no time as an aid for making decisions.

It should be emphasized that the computer code of EMIT is available to the scientific community via the Github hosting platform: https://github.com/teddy4445/epidemiological_social_sim (accessed on 30 October 2022). Of note, as the Twitter data are confidential, we deploy the trained model based on the data, thus the original data are available on-demand for expansion.

EMIT is also an efficient pandemic management decision-support tool, made from a time perspective. Indeed, at any time, and more particularly during a crisis such as a pandemic, social media are highly reactive. Accordingly, EMIT provides results in a short time. The EMIT’s algorithmic complexity allows relatively high-speed computation, as the utilization of the next wave predictor is of O(1) and the social-PIP simulator is of O(N) for a population of size *N*.

Nevertheless, EMIT has some current limitations. EMIT has been evaluated during the current COVID-19 pandemic, with social media messages collected from Twitter, in English, and from North America. Accordingly, EMIT efficiency is limited at this time to this context and to the effects of the biological properties of the SARS-CoV-2 to mutate over time. Moreover, the results of this version of EMIR are limited to the US; thus, predictions and simulations have limited spatial resolutions of the waves and PIPs implementation.

To overcome these limitations, future EMIT versions will be adapted to use, in addition to tweets, data from other social media, such as Facebook, Reedit, and LinkedIn. Along the same line, a future collected message will have multilingual sources from multiple geographical locations to enhance the ability of EMIT to predict new waves and to simulate the population compliance to PIPs.

However, using SMNs induces a passive exclusion of nonusers or passive users (only reading and not posting) of these communication channels [23].

Another limitation of the EMIT model is its sensitivity to the definition of “beginning of a new wave”. In other terms, this means that the “beginning of a new wave” must be defined by looking at a few parameters together and the changes in trends of specific topics published on social networks [23], like the trends of numbers of exposed, ill, and dead people from spreading disease, and both (i.e., trends of topics, trends of causalities) in a specific geographical area.

From a technical perspective, the current version of EMIT is available as a computational code package; in the future, EMIT could have a user interface and be hosted on a website to be used more freely by the scientific and medical communities.

## 5. Conclusions

Endemic, epidemic and pandemic waves are well-known phenomena that challenge the healthcare system and actually the society as a whole.

### 5.1. Implications for Healthcare Practice

Policymakers can take advantage of the dynamics in SMNs to identify, manage and control the disease spread. In order to effectively do so, healthcare professionals need reliable tools to predict ahead of time pandemic waves. Moreover, the practical implementation of the pandemic intervention policies is considerable in a fully interconnected world wherein the population’s willingness changes over time and under outer influences, such as social media.

To tackle these challenges, in this study we developed a social-epidemiological tool, called EMIT, that allows policymakers to both predict the next pandemic and estimate in advance the impact of multiple health communication intervention strategies. Thus, our tool enables us to exploit the pandemics’ unique behavior. To study the performance and limitations, we implemented our tool for the COVID-19 pandemic in the US, demonstrating promising results.

In our analyses, we reveal an important outcome for policymakers, documenting that the duration of the delay between the prediction and the actual beginning of the pandemic is strongly related to the bio-clinical properties of the pathogen. As such, our tool is better used in the second or even third wave of an epidemic/pandemic, when sufficient data are available.

### 5.2. Future Research Directions

Epidemics and pandemics are not limited to communicable diseases such as COVID-19 or Influenza. Future versions of EMIT will be adapted to track and anticipate waves of massive and recurrent behavioral abnormalities (such as addiction to some games) in social media, and to enable decision-makers to define counter-measures for limiting the effects of these behaviors [53].

## Figures and Tables

**Figure 1 ijerph-19-16023-f001:**
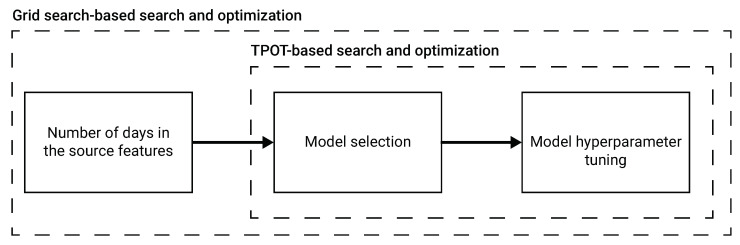
A schematic view of the auto-ML process to obtain the *next wave* prediction model. The number of days in the source features is picked using a grid search based approach, aiming to optimize the $F_{1}$ score of the obtained model. Based on the picked number of days in the source features, we utilize TPOT [27] to find an ensemble of ML regression models and their hyperparameters.

**Figure 2 ijerph-19-16023-f002:**
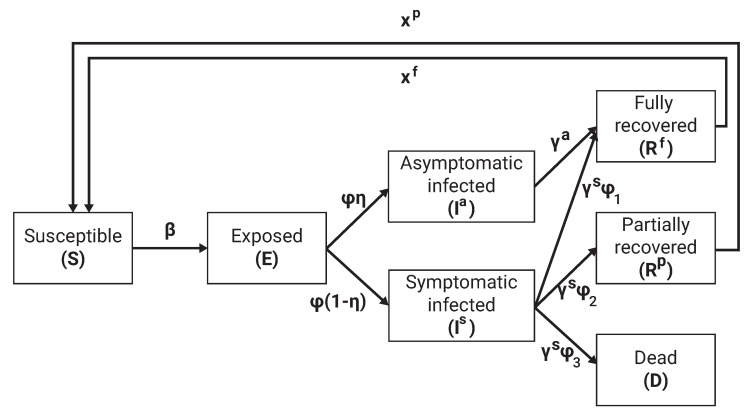
Schematic view of transition between epidemiological stages.

**Figure 3 ijerph-19-16023-f003:**
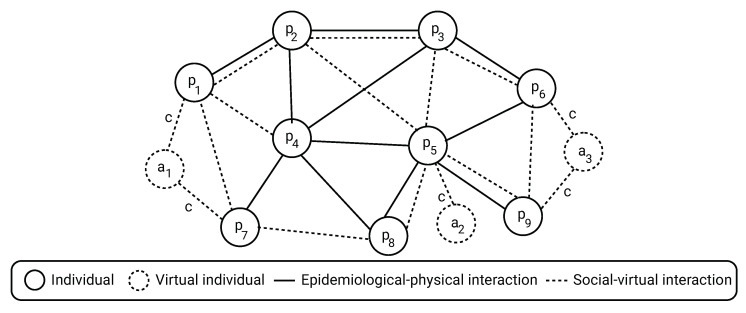
Schematic view of the model with both individuals in the populations and virtual individuals
set by a CP to influence on the PIPs the population performs.

**Figure 4 ijerph-19-16023-f004:**
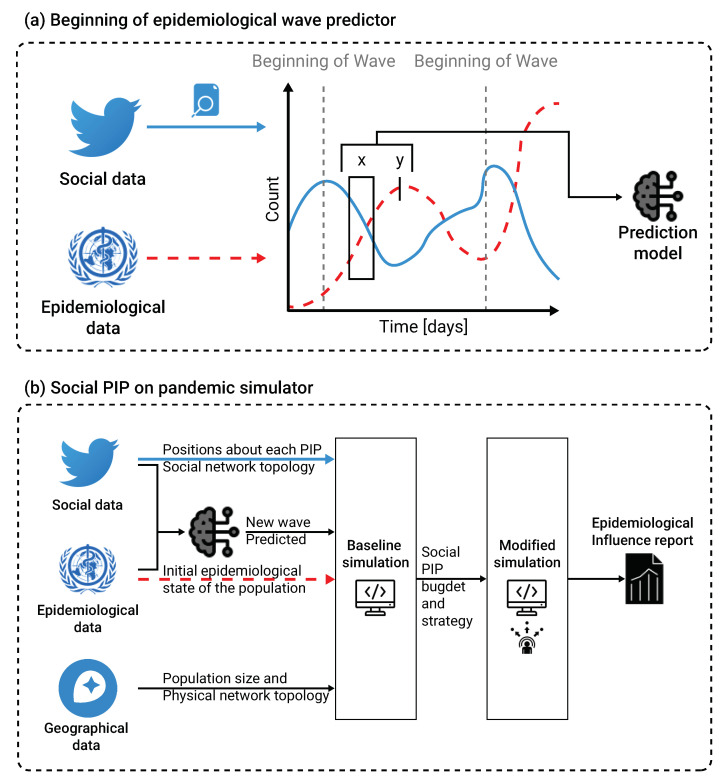
(**a**) a schematic view of the epidemiological wave predictor training process is presented. Namely, we used epidemiological data from WHO [4] and social media data from Twitter, and the
natural language processing method to convert the information into time-series data. Using these data, we trained an XGboost model [29] that obtained a moving window on the epidemiological and social data as the input data and binary signal indicating if a beginning of a new epidemiological wave occurs. The solid (blue) line represents the social data while the dashed (red) line represents the epidemiological data. A fixed delay between the input and target data is provided to the model
by the user. (**b**) a schematic view of the social PIP’s evaluation is presented. In order to use the
spatial component of the proposed extended SIR model, geographical and sociological data in the
form of population size and physical network topology are introduced. Based on these data, a
baseline simulation was computed. Afterward, the user defines a social PIP strategy and budget
and re-computes the modified simulation. EMIT reports the differences between the two runs as the
influence of the social PIP.

**Figure 5 ijerph-19-16023-f005:**
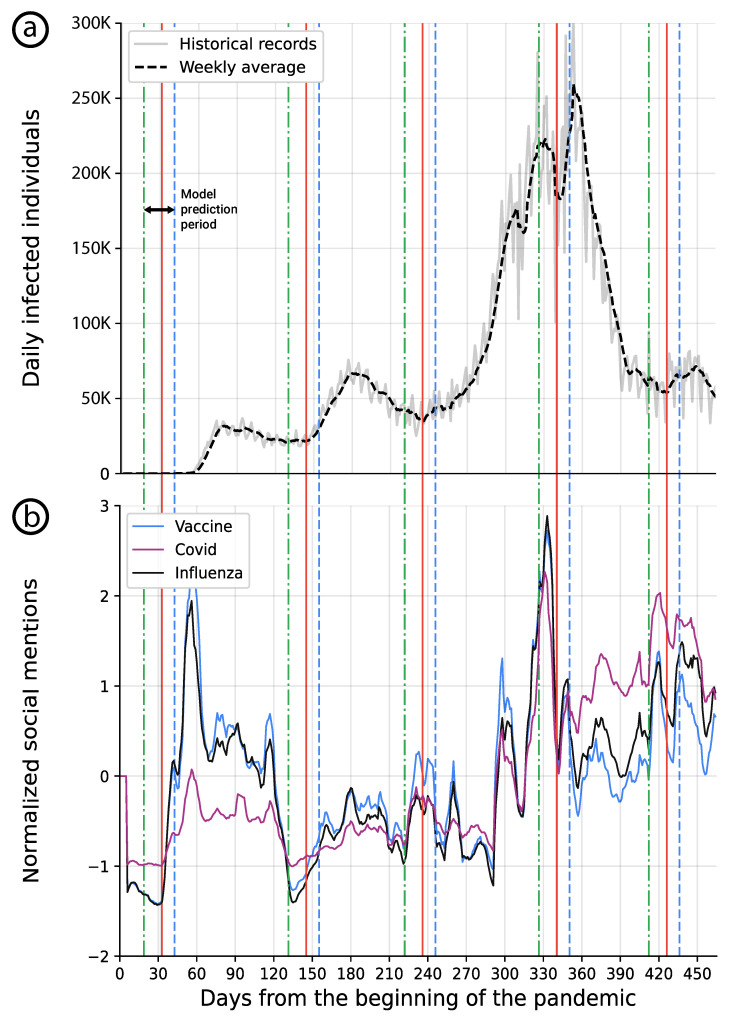
(**a**) the daily official number of infected individuals in the USA is shown in a solid gray line with a weekly moving average line shown in dashed black. (**b**) the weekly average z-score of a normalized number of Twitter posts that were related to the “vaccine”, “covid”, and “influenza” subjects are presented, respectively. The blue dashed line indicates the clinically identified beginning of a wave, the red solid line indicates the retrospective beginning of a wave, and the green dotted line indicates the two-week model’s prediction date.

**Figure 6 ijerph-19-16023-f006:**
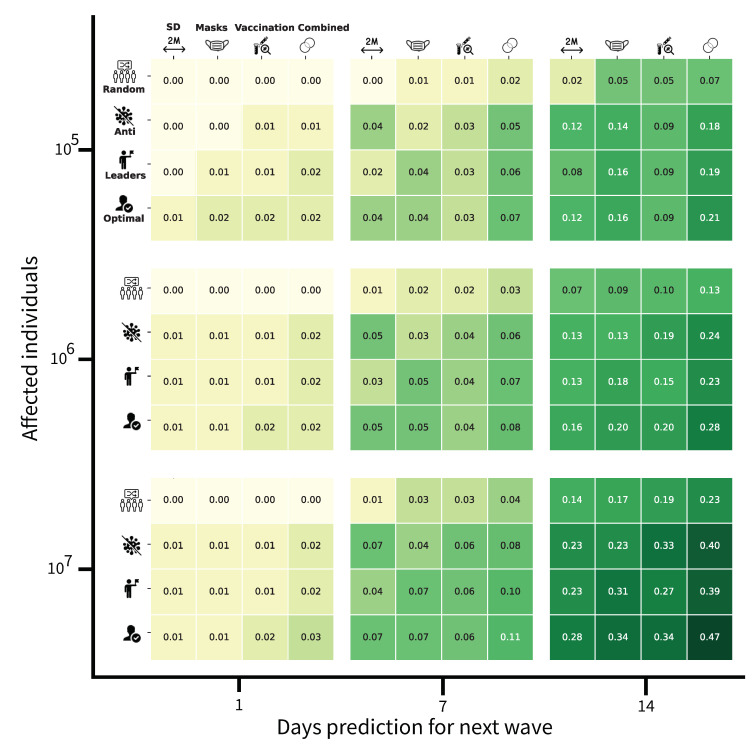
Four-dimensional analysis of the social PIP. The first dimension is the delay between a prediction and the beginning of an epidemiological wave. The second is the number of affected individuals, as a function of the PIP’s budget. The third dimension relates to the PIP which is promoted by the social PIP. Forth is the social PIP’s strategy. The non-social PIPs are social distancing (SD), mask-wearing, vaccination, and all three (combined). The social PIP strategies include advertising to a random subset of the population (Random), advertising to the individuals with the most anti-PIP point of view (Anti), advertising to the most socially-connected individuals (Leaders), and advertising to the optimal subgroup of the population assuming an all-know government (Optimal). The values correspond to the average reduction in the basic reproduction number ($R_{0}$) in the two weeks after the next epidemiological wave has begun. The green color gradient is proportional to the value of the ($R_{0}$) reduction, namely, darker is a higher value. The results shown are computed for the case of COVID-19 in the US based on data from 1 March 2020 to 31 April 2021.

## Data Availability

The epidemiological data used in the research is available online while the social data is available upon writing request from the authors.

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
