# Peer review of "Early Detection and Control of the Next Epidemic Wave Using Health Communications: Development of an Artificial Intelligence-Based Tool and Its Validation on COVID-19 Data from the US"

_ijerph, 2022, doi:10.3390/ijerph192316023_

Round 1

Reviewer 1 Report

The authors have presented a model with a computational tool, called EMIT (Epidemic and Media Impact Tool), for recognizing pandemic waves mathematically, using topics of relevance on social media networks and outbreak spread. The topic is quite new and very interesting. The mathematical contribution is not bad. The paper is quite well-written and easy to read, although some typos are found. It should be revised again.

However, a schematic view of the epidemiological wave predictor training process is not so clear. It should be elaborated more clearly. In addition, some literature reviews for artificial intelligence and related technologies for COVID-19 should be seriously added. For example, the following recent papers using artificial intelligence and extended reality for COVID-19 should be mathematically added and discussed:

1.Monitoring COVID-19 Disease Using Big Data and Artificial Intelligence-Driven Tools.

2.Digital Transformation and Emerging Technologies for Fighting COVID-19 Pandemic 2021: 163-174,

3.Roles of Artificial Intelligence and Extended Reality Development in the Post-COVID-19 Era. HCI (41) 2021: 445-454,

4.Novel coronavirus (COVID-19) diagnosis using computer vision and artificial intelligence techniques: a review. Multim. Tools Appl. 80(13): 19931-19946 (2021),

5.Exploring the Impacts of COVID-19 on Digital and Metaverse Games. HCI (39) 2022: 561-565,

6.Artificial Intelligence and Machine Learning as a Tool for Combating COVID-19: A Case Study on Health-Tech Start-ups. ICCCNT 2021: 1-5.

Moreover, a four-dimensional analysis of the social Pandemic Intervention Policies should be discussed in detail. In conclusion, the paper could be accepted after proper revision and added more references. If not, I am afraid to reject this paper.

Author Response

Dear Editors and Reviewers,

We are pleased and privileged to re-submit our manuscript “Early Detection and Control of the Next Epidemic Wave using Health Communications: Development of an Artificial Intelligence-based Tool and its Validation on COVID-19 Data from the US“  for your reconsideration to be published in IJERPH.

First, we would like to sincerely thank the reviewers for their detailed and thought-provoking feedback. It was a delight to address the comments and questions of the reviewers and we truly see how these significantly improve the quality of our manuscript. 

Following the reviewers’ comments, we altered the manuscript to better outline the proposed work and to be better suited for a high-standard journal such as IJERPH. In particular, we performed the following improvements:

  1. Added much more references to the paper to support our claims and better review the most recent developments in the field of AI tools used to control COVID-19.
  2. Better explain Figure 5.
  3. Better explain the model’s training process. 

To make the review process smoother, we highlighted all additions and edits in the manuscript in bold font. In addition, we address below, point-by-point, the reviewers' comments, and questions. 

We look forward to hearing from you and hope you will positively consider our manuscript for publication in your journal.

Kind regards,

Teddy Lazebnik*, Svetlana Bunimovich-Mendrazitsky, Shai Ashkenazi, Eugene Levner, Arriel Benis

Reviewer #1:

Comment #1: “The authors have presented a model with a computational tool, called EMIT (Epidemic and Media Impact Tool), for recognizing pandemic waves mathematically, using topics of relevance on social media networks and outbreak spread. The topic is quite new and very interesting. The mathematical contribution is not bad. The paper is quite well-written and easy to read, although some typos are found. It should be revised again.“

Answer #1: We would like to thank the reviewer for the kind words. We asked a native English speaker to carefully review the entire work and fix any typos and badly worded sentences. As such, we hope the new version is grammatical errors- and typos- free. 

Comment #2: “However, a schematic view of the epidemiological wave predictor training process is not so clear. It should be elaborated more clearly.

Answer #2: We would like to thank the reviewer for pointing our attention to this shortcoming. Following this comment, we added the following text to make sure the training of the epidemiological wave predictor is clearer. In addition, we extend the text in the caption of Figure 1 to better convey the figure’s idea better. On top of that, we make the entire code of the project available using GitHub, allowing others to review our model and build on top of it. 

Comment #3: “In addition, some literature reviews for artificial intelligence and related technologies for COVID-19 should be seriously added. For example, the following recent papers using artificial intelligence and extended reality for COVID-19 should be mathematically added and discussed:

1.Monitoring COVID-19 Disease Using Big Data and Artificial Intelligence-Driven Tools. Digital Transformation and Emerging Technologies for Fighting COVID-19 Pandemic 2021: 163-174,

2.Roles of Artificial Intelligence and Extended Reality Development in the Post-COVID-19 Era. HCI (41) 2021: 445-454,

3.Novel coronavirus (COVID-19) diagnosis using computer vision and artificial intelligence techniques: a review. Multim. Tools Appl. 80(13): 19931-19946 (2021),

4.Exploring the Impacts of COVID-19 on Digital and Metaverse Games. HCI (39) 2022: 561-565,

5.Artificial Intelligence and Machine Learning as a Tool for Combating COVID-19: A Case Study on Health-Tech Start-ups. ICCCNT 2021: 1-5.”

Answer #3: We would like to thank the reviewer for sharing with us the most recent academic works in the field of AI tools for COVID-19 control. Following this comment, we introduce these references with some review in the Introduction section. 

Comment #4: “Moreover, a four-dimensional analysis of the social Pandemic Intervention Policies should be discussed in detail.”

Answer #4: Thank you very much for this suggestion. Following this comment, we introduce a new paragraph that better explains the four-dimensional analysis of the social Pandemic Intervention Policies as shown in Figure 6. Shortly, the purpose of the analysis is to find trends in the population size, and affected individuals from the PIP, PIP type, and social media approach. From the figure, it is clear that the number of infected individuals grows but depends on the day's predictions for a new wave. More important, random influence is always less effective than targeting leaders or even anti-PIP individuals. We discuss these points in section 3.3 and their meaning in section 4.

Reviewer 2 Report

The authors use a good data visualization tool to represent the data they get. 

The question is the curves for the three pandemic in Figure 5 is too similar that the data is too perfect.  Authors should  well explain the phenomenon.  

The prediction in the epidemiological wave did not provide the evaluation metric to  gauge its precision, like recall rate, precision rate, etc. 

Author Response

Dear Editors and Reviewers,

We are pleased and privileged to re-submit our manuscript “Early Detection and Control of the Next Epidemic Wave using Health Communications: Development of an Artificial Intelligence-based Tool and its Validation on COVID-19 Data from the US“  for your reconsideration to be published in IJERPH.

First, we would like to sincerely thank the reviewers for their detailed and thought-provoking feedback. It was a delight to address the comments and questions of the reviewers and we truly see how these significantly improve the quality of our manuscript. 

Following the reviewers’ comments, we altered the manuscript to better outline the proposed work and to be better suited for a high-standard journal such as IJERPH. In particular, we performed the following improvements:

  1. Added much more references to the paper to support our claims and better review the most recent developments in the field of AI tools used to control COVID-19.
  2. Better explain Figure 5.
  3. Better explain the model’s training process. 

To make the review process smoother, we highlighted all additions and edits in the manuscript in bold font. In addition, we address below, point-by-point, the reviewers' comments, and questions. 

We look forward to hearing from you and hope you will positively consider our manuscript for publication in your journal.

Kind regards,

Teddy Lazebnik*, Svetlana Bunimovich-Mendrazitsky, Shai Ashkenazi, Eugene Levner, Arriel Benis

Reviewer #2:

Comment #1: “The authors use a good data visualization tool to represent the data they get.”

Answer #1: Thank you very much for the kind words - it is really cheerful to know others like the visualization. 

Comment #2: “The question is the curves for the three pandemic in Figure 5 is too similar that the data is too perfect.  Authors should well explain the phenomenon.”

Answer #2:  Thank you for this question. Figure 5b shows the normalized (using the weekly average z-score) number of mentions on Twitter of three clusters called “Vaccine”, “covid”, and “influenza”. As such, it is not showing the course of the pandemic in 5b and indeed the influence and COVID pandemics are not perfectly aligned. From Figure 5b, one can learn that people discussed (during the sampled time) COVID and influenza together. It is not supervised as a lot of time in the media the two have been compared and a social discussion raise about both more or less together. Following this comment, we first emphasize the fact that Figure 5b is representing social discussion and not the course of the pandemic itself. Moreover, we include the point raised by the reviewer and our response as part of the discussion. 

Comment #3: “The prediction in the epidemiological wave did not provide the evaluation metric to gauge its precision, like recall rate, precision rate, etc.”

Answer #3:  Thank you for this comment. Following this comment, we introduce metrics to gauge its performance as follows: “In particular, for the case of COVID-19 in the US, EMIT obtain an area under the receiver operating characteristic (ROC) curve (AUC) of \(0.909\) and \(F_1\) score of \(0.899\) using data from December 30, 2019, and April 30, 2021.” From this sentence, we show the model’s F_1 score which combines precision and recall using a harmonic mean. Moreover, we provide the AUC score which is commonly used for these kinds of models. On top of that, we provide the code of the project using GitHub for maximum transparency and reproducibility.